

# Regenerative plantlets with improved agronomic characteristics caused by anther culture of tetraploid potato (*Solanum tuberosum* L.)

Li Zhang[1,2], Feng-jie Nie[1], Lei Gong[1], Xiao-yan Gan[1], Guo-hui Zhang[3], Xuan Liu[1], Wen-jing Yang[1], Lei Shi[1], Yu-chao Chen[1], Rui-xia Xie[3], Zhi-qian Guo[3] and Yuxia Song[1]

[1] Research Center of Agricultural Biotechnology, Ningxia Academy of Agricultural and Forestry Sciences, Yinchuan, Ningxia, China
[2] College of Agriculture, Gansu Agricultural University, Lanzhou, Gansu, China
[3] Guyuan Institute of Agricultural Sciences, Ningxia Academy of Agricultural and Forestry Sciences, Guyuan, Ningxia, China

Corresponding authors
Zhi-qian Guo,
nxguozhiqian@126.com
Yuxia Song, songyx666@163.com

## ABSTRACT

**Objective**. As the primary means of plant-induced haploid, anther culture is of great significance in quickly obtaining pure lines and significantly shortening the potato breeding cycle. Nevertheless, the methods of anther culture of tetraploid potato were still not well established.

**Methods**. In this study, 16 potato cultivars (lines) were used for anther culture *in vitro*. The corresponding relation between the different development stages of microspores and the external morphology of buds was investigated. A highly-efficient anther culture system of tetraploid potatoes was established.

**Results**. It was shown in the results that the combined use of 0.5 mg/L 1-Naphthylacetic acid (NAA), 1.0 mg/L 2,4-Dichlorophenoxyacetic acid (2,4-D), and 1.0 mg/L Kinetin (KT) was the ideal choice of hormone pairing for anther callus. Ten of the 16 potato cultivars examined could be induced callus with their respective anthers, and the induction rate ranged from 4.44% to 22.67% using this hormone combination. According to the outcome from the orthogonal design experiments of four kinds of appendages, we found that the medium with sucrose (40 g/L), $AgNO_3$ (30 mg/L), activated carbon (3 g/L), potato extract (200 g/L) had a promotive induction effect on the anther callus. In contrast, adding 1 mg/L Zeatin (ZT) effectively facilitated callus differentiation.

**Conclusion**. Finally, 201 anther culture plantlets were differentiated from 10 potato cultivars. Among these, Qingshu 168 and Ningshu 15 had higher efficiency than anther culture. After identification by flow cytometry and fluorescence *in situ* hybridization, 10 haploid plantlets (5%), 177 tetraploids (88%), and 14 octoploids (7%) were obtained. Some premium anther-cultured plantlets were further selected by morphological and agronomic comparison. Our findings provide important guidance for potato ploidy breeding.

## INTRODUCTION

Potato (*Solanum tuberosum* L.), an important crop worldwide, is native to Peru and the Andean regions of Bolivia (*Spooner et al., 2005*). It is universally grown worldwide due to its advantages of wide adaptability and strong tolerance to arid soil. Since the start of the potato staple food strategy in 2015, the potato industry has developed rapidly and played a vital role in consolidating achievements in poverty alleviation, leading to rural revitalization (*Li et al., 2015*). Most potatoes are autopolyploid species with highly heterozygous genotypes, narrow genetic bases, self-pollinating, and self-incompatibility. Therefore, creating new potato strains with conventional breeding methods is becoming increasingly challenging.

Haploid breeding is a technology in which anther/microspore culture *in vitro* or androgenesis/gynogenesis is used to produce haploids. These haploids form pure diploids by chromosome doubling, which can be used to mine specific trait genes and to study the genetic basis and the ideal population for molecular marker-assisted breeding (*Collard et al., 2005*). However, it can also be conducted by genetic transformation, RNA interference, and gene editing (*Bhowmik et al., 2018*; *Brew-Appiah et al., 2013*; *Wijnker et al., 2012*). More importantly, it is significant to speed up the breeding process and improve breeding efficiency (*Rokka, 2021*; *Song et al., 2005*).

Currently, anther culture is the primary way of haploid induction. During the process of anther culture, male ligand cells or their precursors deviate from the path of their microspore stage, leading to the development of haploid embryos and plants (*Barany et al., 2005*; *Reynolds, 1997*; *Seguí-Simarro, 2010*; *Shariatpanahi & Ahmadi, 2016*). The first adventitious embryoid obtained was derived from potato anther callus but failed in sprout regeneration (*Kohlenbach & Geier, 1972*). Tetraploid and diploid potato anthers were induced. The anther culture ability of some wild potatoes, such as *S. chacoense* Bitt and *S. phureja* Juz., is higher than that of cultivated potatoes (*Dunwell & Sunderland, 1973*; *Foroughi-Wehr et al., 1977*).

Homozygous bihaploids resistant to stem-nematode and antiviral were obtained through anther culture (*Wenzel & Uhrig, 1981*). The regenerative plantlets were induced from the anther of tetraploid cultivated species (2n ± 4x ± 48), and some superior strains were obtained by crossbreeding with the wild species using the excellent dihaploid (2n ± 2x ± 24) (*Dai et al., 1993*). The number of embryoids could be significantly improved by anther pre-culture at a high temperature of 35 °C (*Chani, Veilleux & Boluarte-Medina, 2000*; *Wang, Ran & Dai, 1990*).

Previously, the anther effect of 23 tetraploid potato genotypes was evaluated, and regenerative plantlets were obtained (*Rokka, Pietilä & Pehu, 1996*). Although many steps in the anther culture system have been clearly defined (*Jain, Sopory & Veilleux, 1996*; *Rokka, 2021*), a thorough investigation is required to increase the frequency of embryos and plant regeneration from anther, especially when new varieties or strains are used as the anther donors. In this study, a superior tetraploid anther culture system in potatoes was established. We explored the corresponding relationship between different development stages of microspores and the external morphology of buds, callus induction, plant regeneration, and ploidy identification of regenerative plantlets. Moreover, two cultivars

with high efficiency of anther culture were screened out. These findings guide the theory and practice basis for future potato ploidy breeding.

# MATERIALS AND METHODS

## Plant materials

The 16 potato cultivars (2n = 4x = 32) were planted in the Touying test base in Guyuan Branch, Guanzhuang test base in Longde County, and Jiali company test base in Xiji County, Ningxia Academy of Agriculture and Forestry Sciences. The cultivars were Qingshu 168, Qingshu 9, Favorita, Ningshu 15, Ningshu 16, Ningshu 18, Longshu 7, Longshu 10, Longshu 14, Victoria, Xisen 6, Lishu 6, Zhonghan 37, Atlantic, Huashu 13, and Huashu 9. These cultivars were cultivated from 2017 to 2020.

## Identification of the sampling period and pre-treatment of the sample

During the potato buds and early bloom period, different sizes of buds were collected from the tested plantlets from 9:00 a.m. to 10:00 a.m. daily to measure their lengths. Samples were fixed with Carnoy's Fluid (three ethanol: one glacial acetic acid) for 24 h and were preserved in 70% alcohol and maintained in the refrigerator at 4 °C. Samples were fixed and placed on a glass slide for observation. Improved magenta dye was added to the samples. After that, a cover glass was placed on the samples. The anther development period and the outer morphological traits of the buds were checked under a microscope (Olympus BX-51) and recorded to determine the sampling period. The buds in the mid-uninucleate and late-uninucleate stages were selected to preserve at 4 °C for 48–72 h and washed with running water for 10 min. The buds were sterilized with 70% alcohol for 30 s on an ultra-clean workbench, rinsed twice with sterile water, disinfected with 0.1% $HgCl_2$ for 8 min, and rinsed 4–5 times with sterile water. The anther was peeled from the buds to be sown.

## Anther callus induction

### Treatment with different hormone combinations

The primary medium for an anther callus induction culture consisted of MS (*Murashige & Skoog, 1962*) + sucrose (30 g/L) + agar powder (4 g/L). Eight different hormonal combinations (pH 5.8) were added, as listed in Table 1. Each material was sown in 15 bottles. Each bottle was sown with 10–15 anthers. The experiment was repeated three times. The anthers were cultured in the dark. After pre-treatment for 48 h at 35 ± 2 °C, the anthers were transferred to 24 ± 2 °C. The subculture was conducted for 30 d. Changes in the anther shape and callus growth were regularly observed and recorded. Callus induction rate (after 60 d) = number of anthers produced/number of anther inoculum ×100%. Analysis of variance for the required data was performed using SPSS software. A *P*-value lower than 0.05 was adopted to analyze the differences among treatments.

### Treatment with different appendages and induction culture

Qingshu 168 was set as the sample. Based on successfully identifying the optimal hormone combination and pre-treatment duration of high temperatures, four kinds of appendages (*e.g.*, sucrose, $AgNO_3$, activated carbon, and potato extract) were added to the induction
**Table 1 Treatment with plant hormone combinations in potato anther culture.**

| Treatment | Culture medium | Purpose | Hormone levels (mg/l) | | | | | | | |
|---|---|---|---|---|---|---|---|---|---|---|
| | | | NAA | 2,4-D | 6-BA | KT | GA$_3$ | ZT | IBA | IAA |
| 1 | | | 0 | 0.5 | | 0.5 | | | | |
| 2 | | | 0.5 | 0.5 | | 0.5 | | | | |
| 3 | | | 0.5 | 0.5 | | 1 | | | | |
| 4 | MS | Callus induction | 0.5 | 1 | | 0.5 | | | | |
| 5 | | | 0.5 | 2 | | 0.5 | | | | |
| 6 | | | 0.5 | 2 | | 1 | | | | |
| 7 | | | 0.5 | 1 | | 1 | | | | |
| 8 | | | 0.5 | 1 | | 2 | | | | |
| 9 | | | | | 2 | | 0.5 | | 1 | |
| 10 | MS | Callus differentiation | | | 2 | | 0.5 | | | 1 |
| 11 | | | | | 2 | | 0.5 | 1 | | |
| 12 | | | | | 2 | | 0.5 | 3 | | |

**Notes.**

MS, Murashige and Skoog; NAA, Naphthalene acetic acid; 2,4-D, 2,4-dichlorophenoxyacetic acid; 6-BA, 6-benzylaminopurine; KT, kinetin; GA$_3$, Gibberellin 3; ZT, Zeatin; IBA, Indolebutyric acid; IAA, indoacetic acid.

**Table 2 The design of three levels of different factors for appendages of potato anther culture.**

| Level | Sucrose (g/L) | AgNO$_3$ (mg/L) | Active carbon (g/L) | Potato extract (g/L) |
|---|---|---|---|---|
| 1 | 40 | 0 | 0 | 0 |
| 2 | 60 | 30 | 1.5 | 100 |
| 3 | 80 | 50 | 3 | 200 |

**Table 3 The orthogonal design L9 (3$^4$) of main factors for appendages of potato anther culture.**

| Factors | Treatment | | | | | | | | |
|---|---|---|---|---|---|---|---|---|---|
| | I | II | III | IV | V | VI | VII | VIII | IX |
| Sucrose (A) | 1 | 1 | 1 | 2 | 2 | 2 | 3 | 3 | 3 |
| AgNO$_3$ (B) | 1 | 2 | 3 | 1 | 2 | 3 | 1 | 2 | 3 |
| Active carbon (C) | 1 | 2 | 3 | 2 | 3 | 1 | 3 | 1 | 2 |
| Potato extract (D) | 1 | 2 | 3 | 3 | 1 | 2 | 2 | 3 | 1 |

medium. L$_9$(3$^4$) orthogonal array was constructed. Nine experiments were conducted and repeated three times (Tables 2–3). The culture conditions and data analysis were the same as the treatment with different hormone combinations.

### Differentiation culture and rooting culture

The induced callus was transferred to the MS basic media containing different hormone combinations for differentiation culture with alternate light (16 h) and dark (8 h) at 24 ± 2 °C with 36 μmol/m$^2$/s light radiation. The subculture was conducted once

every 30 d. The differentiation of the callus was regularly observed and recorded. The differentiation rate of callus = (the number of seedlings or plantlets produced/number of callus inoculations ×100%).

When two small pieces of leaves were grown from seedlings differentiated by subculture, the sample was transferred to the medium (1/2 MS + 0.01 mg/L of 1-Naphthylacetic acid (NAA) + 15 g/L of sucrose + 4 g/L of agar powder) for a 30-day root culture. The light and temperature conditions were the same as those in the differentiation culture. Ploidy identification and transplantation were conducted when 7–8 pieces of leaves were grown, and the root was 4–5 cm. The data analysis was the same as the treatment with different hormone combinations.

### Cytological observation of the process of callus differentiation

The callus transferred to the differentiation medium at different times was divided into several parts. The formalin-aceto-alcohol solution in 5:5:90 (v/v/v) concentrations, 38% formaldehyde, glacial acetic acid, and 70% ethanol was used to fix the tissue for 48 h. After that, the tissue was covered with wax and cut into slices of 5 $\mu$m sections. The sections were stained with 0.1% Safranin O and 0.5% Fast Green. The sealed sections were observed using a Nikon microscope, Eclipse Ci-S, and photographed using Nikon NIS-Elements.

## Ploidy identification and morphological observation of anther culture
### Identification with flow cytometry

The leaves of the tetraploid potatoes were set as the sample. Ploidy identification for 201 regenerative plantlets was conducted with flow cytometry. About one $cm^2$ of the leaf was cut from each regenerative plantlet and immersed in 0.4 mL of partechra lysate, and it was then cut into pieces using a blade. After 3 min, the sample was filtered into a small test tube through a 100 $\mu$m filter and then stained with PartecHR-B solution (1.6 mL) for 2 min. The sample was placed in the flow cytometer (Partec's CyFlow® Cube6) for ploidy analysis. The flow cytometric measurement was repeated twice for each sample with at least 10,000 nuclei. The ploidy of the sample can be determined from the relative position of the cell nucleus peak.

### Preparation of chromosomes

The root was cut when the root tip length of the sample reached 2–3 cm. Before the rooted cutting, a centrifugal tube (0.5 mL) with a top hole was prepared and moistened with water before the experiment. The tube was inserted into the ice after the cut root tip was included. The centrifuge tube with the root tip was placed in an inflatable tank filled with 0.9–1.0 MPa of $N_2O$ for 2 h. After that, 90% pre-cold acetic acid was added into the centrifuge tube in ice bath conditions. The ice acetic acid was removed after the tube was fixed for 10 min. The sample was washed twice with $ddH_2O$. The white part of the root tip was sliced with a blade and placed in a centrifuge tube containing 25 $\mu$L of enzyme liquid (cellulase: pectase = 3:1), which was disassembled under water bath conditions at 37 °C for 1 h.

The tip was washed three times with 70% alcohol, fully broken, shaken using the anatomical needle in the remaining alcohol, and centrifugated at 4,000 r/min. The supernatant was discarded, and the remaining liquid was removed as much as possible

by leaving the tube upside down. Depending on the number of root tips, 25–45 μL of ice acetic acid was added to the centrifuge tube. After instantaneous centrifugation, the tube was shaken fully to help the solution mix thoroughly. The clean slide was placed in a pre-prepared moist box at room temperature of around 23 °C. The cell suspension (8 μL) from the centrifuge tube was absorbed and dropped directly onto the slide. The box was immediately covered. Until the cells dispersed, the slides were removed after they were dried. A microscopic examination was carried out under standard optical or differential microscopes to determine the target phase separation to set aside.

### Identification with fluorescence in situ hybridization (FISH)

Nick translation was used to label DNA to be used as a hybridization probe. 5′FAM-dUTP (green) and TAMRA-dUTP (red) were mainly used. Telomeres were a sequence of TTTAGGG 6 synthesized by Shanghai Biotech Company. The 5S rDNA inserted into the plasmid pTa794 was used as a probe (*Gerlach & Dyer, 1980*), and the 18S rDNA was also inserted into the plasmid pBR322 and was used as a probe (*Gerlach & Bedbrook, 1979*). The probe label contained labeled dUTP (4 μL), DNA (1 μg), and double distilled water (16 μL), with 20 μL of the total system and 1 μg of the total amount of DNA. During the labeling process, the probe was kept away from the light. The ingredients were mixed thoroughly, centrifuged, and placed in a water bath at 16 °C for an overnight reaction.

Around 70% of the methylamide solution (50–100 mL) was dropped onto a chromosome-based microslide. Then, the slide was covered. After the cover glass was quickly shaken off, the tissue section was placed in an 80 °C hybrid box for 2 min to denature the chromosomes. The tissue section passed through an increasing concentration (70%, 95%, and 100%) of alcohol baths pre-cold to −20 °C for 5 min to dehydrate. The slide was removed for natural dryness. Then, a hybridization solution was made. Deionized formamide (7.5 μL, 20× SSC 1.5 μL), ssDNA (1.5 μL), B-DNA (1 μL), 50%DS (3 μL), and probe (1 μL) were added in order. After instantaneous centrifugation, the section was placed in a hybrid box at 80 °C and denaturalized for 8–10 min.

The centrifuge tube was quickly removed and placed in the ice-water mixture for more than 5 min. The hybridization solution was dropped on the slide. The cover glass was gently placed to avoid air bubbles and sealed with nail polish. The slide was placed into the grease box and crossbred overnight at 37 °C. The nail polish around the cover glass was removed with tweezers. The slide was placed in a staining cylinder (2× SSC) and shaken gently until the cover glass fell off. The slide was removed and placed in the staining cylinder (2× SSC) for 5 min at 42 °C, followed by 1× PBS for 5 min. The slide was removed for natural dryness. The slide was dropped with anti-fading agent DAPI in the dark and covered with glass. The slides that dealt with *in situ* hybridization were photographed under the DP70 CCD of the Olympus BX70 fluorescence microscope.

## Phenographic observation of anther-cultured plantlets

The seedlings of the anther-breeding and regenerative plantlets were expanded and transplanted into greenhouses. Then, morphological traits, such as plant height and leaf color, and agronomic traits, such as the number of tubers, were regularly observed.

# RESULTS

## Relationship between the external morphology of buds and the microspore development stage

The developmental stage of microspores is an essential factor affecting the male gametophyte culture. Microspores are reported to be more sensitive to the environment and more likely to be induced at the monokaryotic or late-uninucleate stage. By observing the external morphology of flower buds (mainly length and color) and the microspore development stage of 16 potato varieties, the corresponding relationship between them was further obtained.

From the late-uninucleate stage to the binucleate stage, the calyx is about the same length (average length of $4 \pm 1$ mm) as the petal of a potato anther, and the calyx was closely encased by the petal. The outer size and color of the anther exhibit slight differences according to the cultivar, and the color is mainly green and yellow-green (Figs. 1A–1D).

## Effects of different hormone combinations on potato anther callus induction

After being treated with eight kinds of hormone combinations for 10 d, some anthers grew bigger, some browned, and the degree of browning showed significant differences among the different cultivars. White or pale-yellow callus formed at some additional cracks after 30 d. The induction rate of anther callus was significantly different due to the treatment of varying hormone combinations after 60 d (Fig. 2). The combined use of NAA (0.5 mg/L), 2,4-D (2,4-Dichlorophenoxyacetic acid 1.0 mg/L), and KT (Kinetin 1.0 mg/L) was the best recipe of hormone pairing for anther callus. Calluses were induced from 10 cultivars, with an induction rate of 4.44–22.67%. Ningshu 15 had the highest induction rate, followed by Qingshu 168.

## Effects of different appendages on potato anther callus induction

Qingshu 168 was used as the sample. The induced medium with hormone combinations of NAA (0.5 mg/L) + 2,4-D (1.0 mg/L) + KT (1.0 mg/L), and then the orthogonal design L9($3^4$) of four kinds of appendages was conducted. As shown in Table 4, there were significant differences in potato anther callus rates among different combination appendages in potato anther callus. Among them, callus treated with V (26.3%) and VI (21.25%) achieved the highest induction rate (Table 4). According to the visual analysis of the orthogonal design L9($3^4$) of different factors for appendages, the combination of 60 g/L of sucrose, 30 mg/L of AgNO$_3$, 1.5 g/L of activated carbon, and 200 g/L of potato extract was the best for the induction of anther callus. The potato anther culture appendage scheme was obtained with optimal levels of various factors. According to the Margin R, the order of effect of each factor was A >C >D >B, *i.e.,* the sucrose treatment had the most significant effect on the induction rate, while the treatment of AgNO$_3$ had the slightest impact (Table 5).

## Effects of different hormone ratio on potato differentiation and seeding

The induced callus was placed in a differentiated medium containing four hormone combinations. After 5–10 d, the callus color changed from milky white to light green. After

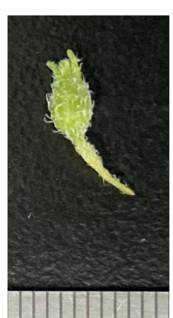

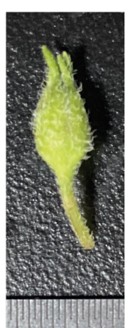

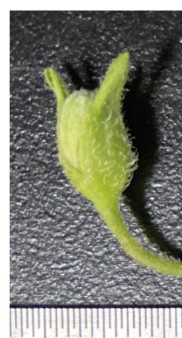

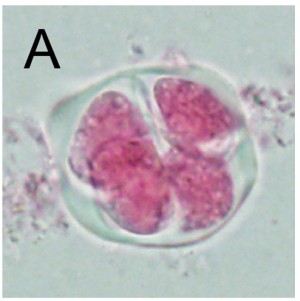

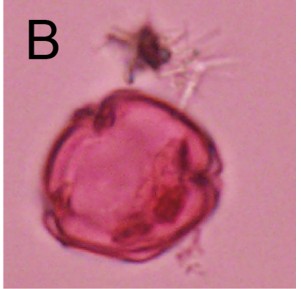

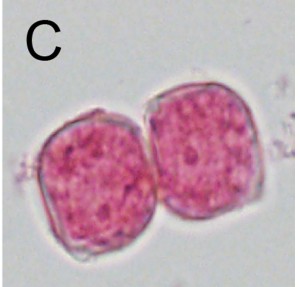

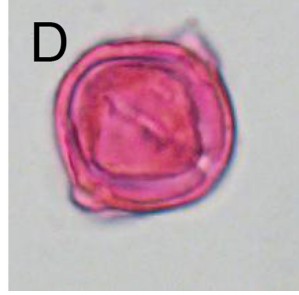

**Figure 1 Correlation of bud size, morphological characteristics, and microspore development stages (Qingshu 168 was set as the samlpe).** (A) Tetrad stage. (B) Mid-uninucleate stage. (C) Late-uninucleate. (D) stage maturity. Scare Bar = 5 μm.

40 d, the green buds were differentiated one after another. After 80 d, in the two hormone combination treatments with Zeatin (ZT) added, callus from 10 cultivars could determine the intact seedlings (Figs. 3A–3D). Among them, hormone combination treatment with GA 0.5 mg/L + 6-BA 2 mg/L + ZT 1mg/L had the highest differentiation rate of 85.7%

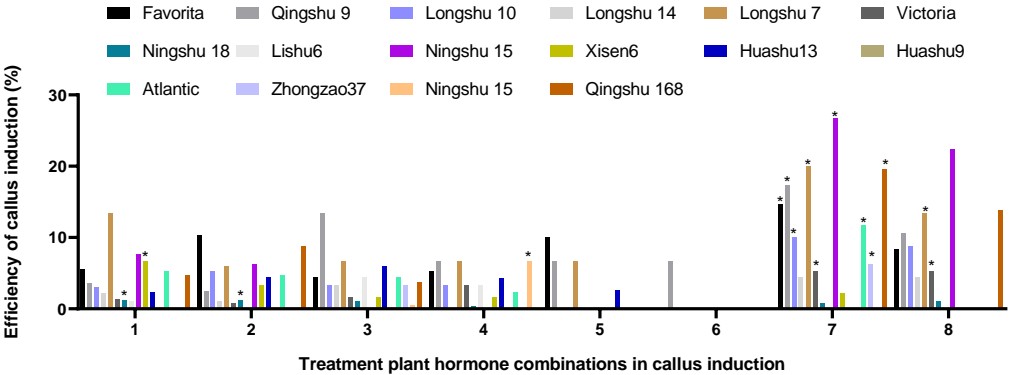

**Figure 2** Effects of different hormone combinations on potato anther callus induction. Note: an asterisk (*) means significant level among induction rate among different cultivars (lines) was at 5%, $P < 0.05$.

**Table 4** Effects of different combinations of appendages on anther callus induction of Qingshu 168.

| Appendage treatment | No.of explants | | | | No.of calli | | | | Average induction rate (%) |
|---|---|---|---|---|---|---|---|---|---|
| | I | II | III | IV | I | II | III | IV | |
| I | 20 | 20 | 20 | 20 | 1 | 3 | 2 | 0 | $7.5 \pm 0.06^{Bb}$ |
| II | 20 | 20 | 20 | 20 | 0 | 3 | 2 | 3 | $10.0 \pm 0.07^{Bb}$ |
| III | 20 | 20 | 20 | 20 | 2 | 2 | 2 | 3 | $11.25 \pm 0.03^{Bb}$ |
| IV | 20 | 20 | 20 | 20 | 1 | 2 | 2 | 0 | $6.25 \pm 0.05^{Bb}$ |
| V | 20 | 20 | 20 | 20 | 5 | 5 | 5 | 6 | $26.3 \pm 0.03^{Aa}$ |
| VI | 20 | 20 | 20 | 20 | 4 | 4 | 5 | 4 | $21.25 \pm 0.03^{Aa}$ |
| VII | 20 | 20 | 20 | 20 | 2 | 3 | 1 | 1 | $8.75 \pm 0.05^{Bb}$ |
| VIII | 20 | 20 | 20 | 20 | 0 | 0 | 2 | 1 | $3.75 \pm 0.05^{Bb}$ |
| IX | 20 | 20 | 20 | 20 | 0 | 2 | 1 | 1 | $5.0 \pm 0.04^{Bb}$ |

Notes.
$P < 0.05$ means at 5% level of significance; $P < 0.01$ means at 1% level of significance.

**Table 5** The visual analysis of the orthogonal design L9 (34) of different factors for appendages of potato anther culture.

| Factor | Sum | | | Averege | | | Minimum | Maximum | Margin R |
|---|---|---|---|---|---|---|---|---|---|
| | Level 1 | Level 2 | Level 3 | Level 1 | Level 2 | Level 3 | | | |
| Sucrose (A) | 0.288 | 0.525 | 0.175 | 0.096 | 0.175 | 0.058 | 0.175 | 0.058 | 0.117 |
| AgNO₃(B) | 0.225 | 0.388 | 0.375 | 0.075 | 0.129 | 0.125 | 0.129 | 0.075 | 0.054 |
| Active carbon (C) | 0.325 | 0.213 | 0.45 | 0.108 | 0.071 | 0.15 | 0.15 | 0.071 | 0.079 |
| Potato extract (D) | 0.375 | 0.4 | 0.213 | 0.125 | 0.133 | 0.071 | 0.133 | 0.071 | 0.063 |

(Table 6). Ningshu 15 and Qingshu 168 had the highest differentiation rates, demonstrating that adding ZT could significantly promote the differentiation of potato anther callus. In addition, the time and type of embryonic callus information were clarified by cytological observation of the differentiation process of the anther callus of Qingshu 168 (Figs. 3E–3G).

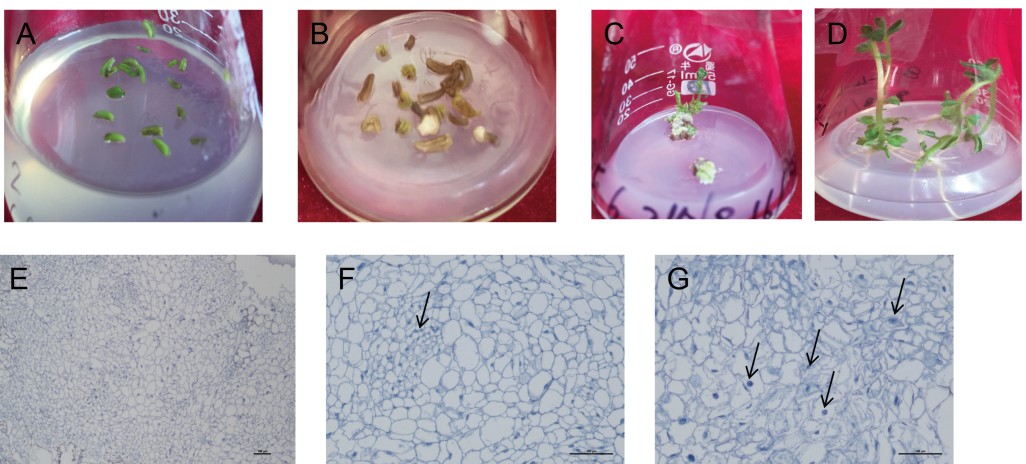

**Figure 3   Induction and differentiation of potato anther callus.** (A) Anther of potato; (B) induction of anther callus; (C) differentiation of anther callus; bar = five mm. (D) Seedling emergence; (E–G) Cytological observation of anther callus differentiation. Arrows refer to embryonic cells. Bar = 100 μm.

As a result, Ningshu 15 and Qingshu 168 were selected as cultivars with a high ability of anther culture.

## Ploidy identification of regenerative plantlets
### Analysis of flow cytometry
Ploidy identification of potato anther regenerative plantlets (Figs. 4A–4F) was conducted with flow cytometry. It was shown by DNA content detection of calli and leaves that the mixed haploid of anther culture materials was determined during the period of callus differentiation. The regenerative plantlets mainly coexisted with diploids, tetraploids, and octoploids (Figs. 4G–4I). Flow cytometry was used to measure the ploidy levels of 201 regenerative plantlets. Among them, 10 were haploid plantlets (5%), 177 were tetraploid (88%), and 14 were octoploid (7%).

### Analysis of chromosomal fluorescence in situ hybridization
DAPI staining for fluorescence *in situ* hybridization was applied to further characterize the cell ploidy of plantlets. The number of chromosomes of anther-cultured plantlets was determined by calculating the number of chromosomes in cells during the metaphase of mitosis of the root tip from at least 20 images (Figs. 4J–4R). The number of chromosomes in the diploid plantlets was 24, with a length of 0.8–2.0 μm, mainly in the middle of the filament chromosome with the small genome. The number of chromosomes in the tetraploid and octoploid plantlets was 48 and 92, respectively, with a length of 0.8–2.0 μm, mainly in the middle and end of the filament chromosomes, with a medium genome. Fluorescent *in situ* hybridization of the above samples was conducted with telomere probes for repeat sequence. Some pericentromeric regions of the chromosome had strong telomere signals, which clarified the number of chromosomes mentioned above, and was consistent with the flow cytometer analysis results.

**Table 6  Effects of different hormone combinations on the differentiation and emergence of potato anther callus.**

| Treatment of plant hormone combinations | | 11 | 12 |
|---|---|---|---|
| Favorita | Differentiation rate (%) | 24.4[c] | 18.2[d] |
| | Emergence of shoots (n) | 30 | 12 |
| Qingshu 9 | Differentiation rate (%) | 9.6[e] | 4.5[f] |
| | Emergence of seeding (n) | 17 | 4 |
| Ningshu 15 | Differentiation rate (%) | 85.7[a] | 65.2[a] |
| | Emergence of seeding (n) | 55 | 20d |
| Qingshu 168 | Differentiation rate (%) | 66.7[b] | 55.5[b] |
| | Emergence of seeding (n) | 25 | 8 |
| Victoria | Differentiation rate (%) | 13.9[d] | 9.5[e] |
| | Emergence of seeding (n) | 13 | 5 |
| Xisen6 | Differentiation rate (%) | 5.0[f] | |
| | Emergence of seeding (n) | 3 | |
| Ningshu 16 | Differentiation rate (%) | 66.7[b] | 46.7[c] |
| | Emergence of seeding (n) | 7 | 2 |
| Zhonghan37 | Differentiation rate (%) | 20.0[d] | 9.1[e] |
| | Emergence of seeding (n) | 2 | |
| Atlantic | Differentiation rate (%) | 5.6[f] | |
| | Emergence of seeding (n) | 3 | |
| Ningshu 18 | Differentiation rate (%) | 27.3[c] | 15.2[de] |
| | Emergence of seeding (n) | 3 | 1 |

## Karyotypic analysis

Fluorescent *in situ* hybridization of samples with different chromosome numbers was performed using 5S rDNA and 18S rDNA repeat sequence probes, respectively (Figs. 4J–4R). The diploid, tetraploid, and octoploid cells identified above had 2, 4, and 8 pairs of chromosomes, respectively, showing strong 5S rDNA (red) and 18S rDNA (green) hybridization signals. It was further determined that the karyotypes of the anther culture plantlets were $2n = 2x = 24$, $2n = 4x = 48$, and $2n = 8x = 94$ (octoploid plants were aneuploidy).

## Identification of morphology and agronomic traits

Significant differences in plant height, leaf color, epidermis, and roots were found through the morphology of the different regenerative plantlets (Figs. 4A–4F). Compared with tetraploids, the diploids had significantly inhibited height and crown diameter, fewer roots, and about a 30-day shorter reproductive period (Figs. 5A–5F). At the same time, the octoploid plantlets had stout stems, thickened leaves, thick leaf epidermis, stubby roots, and slow growth. An agronomic traits survey was thoroughly conducted for 201 anther-cultured plantlets. Compared with the anther receptors, some anthers showed a significant difference in the number of tubers, flesh color, and other indicators (Figs. 6A–6O, Table 7). We screened 23 strains as superior materials for hybrid breeding. Among the analyzed traits, the weight of tubers per plant (F6, F1–1, N3–1, Q93–6, Q92–6, Q12–2, Q14–5, *etc.*) and the number of tubers (Q1–5, F 6–4, Q3–5, Q93–5, *etc.*) increased significantly. In
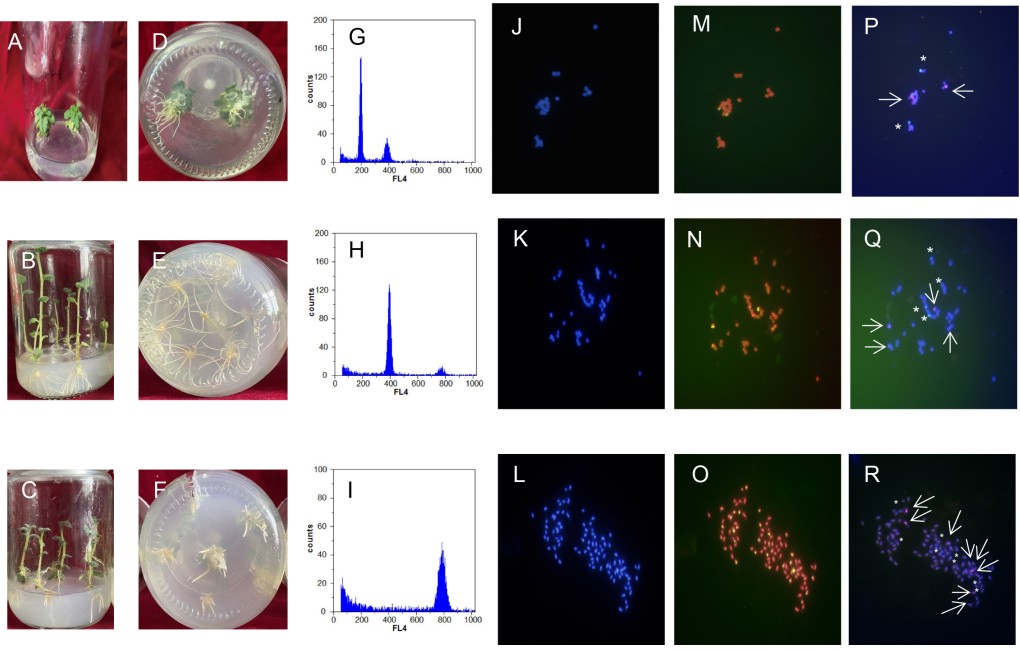

**Figure 4 Comparison of regenerated plantlets with different ploidy levels in culture.** (A–F) Map of DNA content distribution of regenerated plant cells; (G–I) DAPI staining for fluorescence *in situ* hybridization; (J–R) telomeres probes for repeat sequence. A, D, G, J, M, and P: Dihaploid plantlets, 2n = 2x = 24; B, E, H, K, N, and Q: tetraploid plantlets, 2n = 4x = 48; C, F, I, L, O, and R: octaploid mixoploid plantlet, 2n = 8x = 92. Note: Green fluorescence in (M–O) indicates the *in situ* hybridization signal of the telomeric repeats. The arrows and asterisks in (P–R) indicate *in situ* hybridization of 5S rDNA and 18S rDNA, respectively.

addition, the flesh color changed (N7–8, *etc.*), and the large tuber rate was higher (N7–8, F6, *etc*).

## DISCUSSION

### The development period of microspores and the efficiency of anther culture

The development of microspores generally goes through tetrad, mid-uninucleate, late-uninucleate, and binucleate stages and maturity. The appropriate development period of microspores is vital to improving anther callus and embryoid induction. In principle, pollen is sensitive to external stimulation in the mid-uninucleate and late-uninucleate stages, which may be related to changes in the endogenous hormone balance of anther during microspore development. The development period of microspores is closely associated with the efficiency of anther culture. For most plantlets, the late-uninucleate stage is the most suitable. In addition, the external morphology of buds is closely related to the development stage of microspores. Bud length was commonly used as a criterion for external morphological indicators of the development period. In a study by Regalado (2016) about the relationship between different anther sizes of asparagus and the development period, it was shown that there was a significant correlation between them. This can be

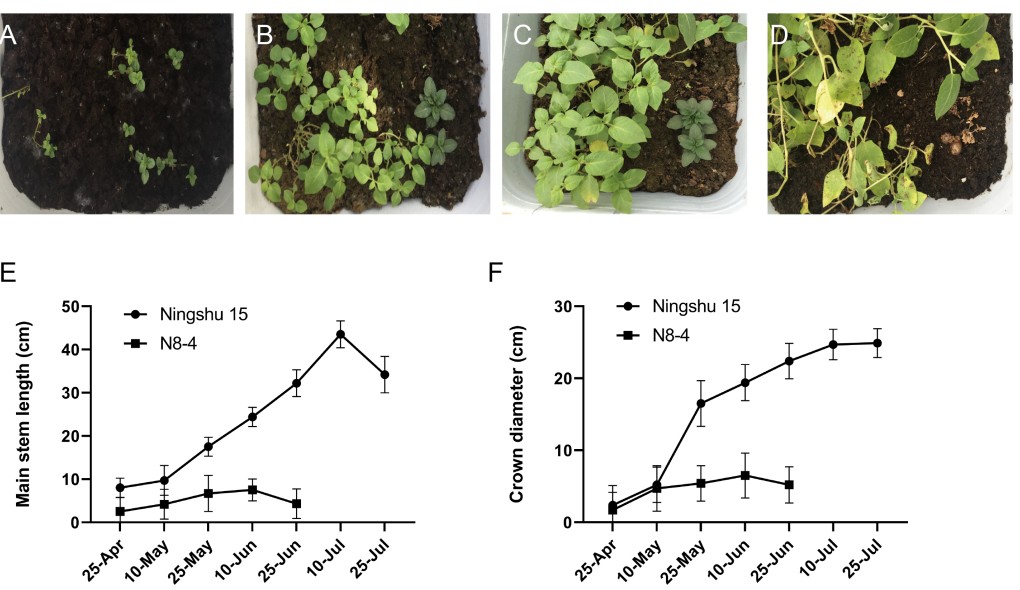

**Figure 5** **Comparison of growth between dihaploid and tetraploid plantlets after transplanting.** (A–D) Growth of tetraploid plantlets (left) and dihaploid plantlets (right) at 3d, 15d, 30d and 60d after transplanting, respectively. (E) Comparison of plant height between tetraploid (Ningshu 15) and dihaploid (N8-4). (F) Comparison of crown diameter between tetraploid (Ningshu 15) and dihaploid (N8-4).

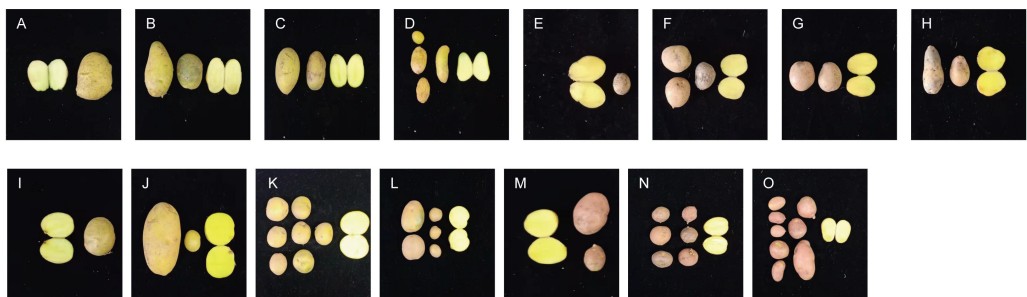

**Figure 6** **Comparison of yield per plant between anther-cultured plantlets and receptor Potatoes.** (A) Favorita; (B–D) F-6 (tetraploid), F2-4 (tetraploid), F3-10 (diploid); (E) Qingshu168; (F–H) Q14-5 (tetraploid), Q12-2 (tetraploid), Q14-2 (octaploid); (I) Ningshu 15; (J–L): N7-8 (tetraploid), N3-2 (tetraploid), N6-3 (tetraploid); (M) Qingshu 9; N,O: Q3-6 (tetraploid), Q2-2 (tetraploid).

used as an indicator for sampling from the appropriate development period and the size of buds and anthers. In this study, the relationship between the development stage of microspores and bud morphology was examined. It was found that these two had close correlations, which could further support that it was feasible to judge the development period of microspores by the outer morphological traits of buds, which can be used as the sampling criteria for potato anther culture.

**Table 7  Statistical analysis of theagronomic traits of some anther-cultured plantlets.**

| No. | Anther culture materials | No. of regenerative plants | Weight of tubers per plant (g) | No. of tubers per plant | Ploidy |
|---|---|---|---|---|---|
| 1 | Favorita (F) | 12 | 40.0 ± 4.20$^{Ee}$ | 2.3 ± 0.32$^{Cc}$ | 4x |
| 2 | F-6 | 12 | 158.3 ± 8.48$^{Aa}$ | 3.3 ± 0.20$^{Aa}$ | 4x |
| 3 | F1-1 | 12 | 156.7 ± 4.24$^{Aa}$ | 2.5 ± 0.20$^{Bb}$ | 4x |
| 4 | F1-2 | 12 | 66.7 ± 4.24$^{Cc}$ | 1.5 ± 0.30$^{Dd}$ | 4x |
| 5 | F4-4 | 12 | 55.0 ± 4.00$^{Dd}$ | 2.2 ± 0.18$^{Cc}$ | 4x |
| 6 | F2-4 | 12 | 81.7 ± 6.97$^{Bb}$ | 2.8 ± 0.20$^{Bb}$ | 4x |
| 7 | F4-6 | 12 | 22.2+2.71$^{Ff}$ | 1.0 ± 0.10$^{Ee}$ | 8x |
| 8 | F3-1 | 12 | 5.8 ± 1.86$^{Gg}$ | 1.0 ± 0.10$^{Ee}$ | 8x |
| 9 | F3-10 | 12 | 1.67 ± 0.50$^{Gg}$ | 0.75 ± 0.05$^{Ee}$ | 2x |
| 10 | Ningshu 15 (N) | 12 | 45.0 ± 2.19$^{Gg}$ | 2.2 ± 0.24$^{Ee}$ | 4x |
| 11 | N3-1 | 12 | 172.1 ± 4.00$^{Aa}$ | 6.8 ± 0.30$^{Cc}$ | 4x |
| 12 | N6-3 | 12 | 154.2 ± 2.71$^{Bb}$ | 7.9 ± 0.30$^{BCb}$ | 4x |
| 13 | N1-7 | 12 | 112.5 ± 4.00$^{Dd}$ | 3.3 ± 0.50$^{Ee}$ | 4x |
| 14 | N7-8 | 12 | 119.2 ± 2.20$^{CDc}$ | 2.6 ± 0.28$^{Ee}$ | 4x |
| 15 | N6-4 | 12 | 65.8 ± 2.71$^{Ff}$ | 12.3 ± 0.20$^{Aa}$ | 4x |
| 16 | N3-2 | 12 | 121.8 ± 1.14$^{Cc}$ | 8.3 ± 1.00$^{Bb}$ | 4x |
| 17 | N1-17 | 12 | 105.0 ± 1.14$^{Ee}$ | 5.2 ± 0.70$^{De}$ | 4x |
| 18 | N8-4 | 12 | 12.0 ± 6.00$^{Hh}$ | 2.5 ± 0.50$^{Ee}$ | 2x |
| 19 | Qingshu (Q9) | 12 | 31.7 ± 5.87$^{Dd}$ | 2.8 ± 0.50$^{Dd}$ | 4x |
| 20 | Q92-2 | 12 | 110.0 ± 5.23$^{BCb}$ | 7.3 ± 0.32$^{BCb}$ | 4x |
| 21 | Q93-6 | 12 | 133.3 ± 5.14$^{Aa}$ | 5.8 ± 0.30$^{Cbc}$ | 4x |
| 22 | Q92-9 | 12 | 92.5 ± 2.00$^{Cc}$ | 9.0 ± 0.90$^{ABa}$ | 4x |
| 23 | Q93-5 | 12 | 107.5 ± 3.30$^{Cb}$ | 10.8 ± 1.00$^{Aa}$ | 4x |
| 24 | Q92-6 | 12 | 125.0 ± 5.00$^{ABa}$ | 4.8 ± 2.00$^{CDc}$ | 4x |
| 25 | Qingshu 168 (Q1) | 12 | 35.0 ± 1.41$^{Cd}$ | 18.3 ± 0.68$^{Bbc}$ | 4x |
| 26 | Q12-2 | 12 | 66.7 ± 2.20$^{Aa}$ | 20.0 ± 1.00$^{Bbc}$ | 4x |
| 27 | Q14-5 | 12 | 65.0 ± 2.34$^{Aa}$ | 23.6 ± 5.00$^{Bb}$ | 4x |
| 28 | Q14-6 | 12 | 45.0 ± 5.00$^{Bbc}$ | 14.6 ± 1.02$^{Bc}$ | 4x |
| 29 | Q13-3 | 12 | 47.5 ± 4.30$^{Bb}$ | 20.4 ± 2.00$^{Bbc}$ | 4x |
| 30 | Q14-7 | 12 | 26.7 ± 3.20$^{Dd}$ | 16.0 ± 2.00$^{Bc}$ | 4x |
| 31 | Q1-5 | 12 | 40.8 ± 4.38$^{BCc}$ | 61.3 ± 2.00$^{Aa}$ | 4x |
| 32 | Q14-2 | 12 | 13.3 ± 2.00$^{Ed}$ | 4.2 ± 6.00$^{Cd}$ | 8x |

## Genotype and flower anther efficiency

Genotypes play an essential role in pollen culture response as a vital endogenous factor. There were significant differences in the response rate of anther culture between different cultivars. Non-response, medium response, and high response are presented according to the potato anther culture ability (*Taylor & Veilleux, 1992*). *Dunwell (2010)* believed the proportion of anther culture that produces microspore embryoids and the regeneration of each anther were independent and controlled by genotypes. The different genotype samples of *Solanum tuberosum L.* were induced with anther culture by *Foroughi-Wehr,*

*Friedt & Wenzel (1982)*, which obtained a plant regeneration rate of 0–5.1%. In the study by *Rokka, Pietilä & Pehu (1996)*, anther culture was conducted in 48 tetraploid potato genotypes, and only 23 genotypes acquired regenerative plantlets. In this study, there was a significant difference in the induction rate of anther callus in 16 potato cultivars. Ten potato cultivars were able to induce callus and obtain regenerative plantlets. Two cultivars with a high response to anther culture were screened out, further indicating that genotype is the most critical factor affecting anther culture.

## Exogenous hormones and potato anther culture

It has been stated in many experiments that in the de-differentiation phase of the anther, exogenous growth hormone 2,4-D is a necessary condition to initiate the division of microspores to form callus (*Cui et al., 2000*). *Dai et al. (1993)* used common cultivars as samples and added 2,4-D (1 mg/L), NAA (2 mg/L), and KT (0.5 mg/L) into MS medium and the callus was induced from the 20,720 sown anthers, with a total induction rate of 1.11%. The effect of 2,4-D on the culture of F1 substitute anther in the pepper hybrid was evaluated, and it was found that 2,4-D treatment significantly increased the number of embryos and plantlets (*Nowaczyk, Nowaczyk & Olszewska, 2016*). In this study, we found that the appropriate concentration of 2,4-D was the key to the induction of the anther callus, which combined 2,4-D (1 mg/L) with KT (1 mg/L) had significantly promoted effects. Low concentrations of ZT are necessary for promoting the differentiation of potato anther callus. The first generation of obtained anther callus from culture was transferred to a media with a specific concentration of ZT, indole-3-acetic acid (IAA), and Gibberellic acid (GA). Green buds were differentiated from callus in 10 cultivars, with a 9.6–85.7% differentiation rate. Histocytological observation of the callus treated with the hormone combinations was conducted. It was revealed that the cells with embryonic callus had commonalities, *i.e.,* small cell size, thick cytoplasm, large nucleus, slight or no liquid bubbles, and divisive solid ability. The induction rate of potato anther embryoids was significantly improved by the results in the research mentioned above. This result was of great significance in enhancing the potato anther embryoid regeneration system and establishing the high-frequency regeneration receptor system of anther embryoids.

## Appendages and potato anther culture

It is reported that $AgNO_3$, activated carbon, and other appendages have specific auxiliary effects on increasing the formation rate of anther embryoid (*Biddington, Sutherland & Robinson, 1988*; *Kim, Lee & Na, 2020*; *Luz et al., 1999*). *Ran & Dai (1993)* added 50–100 µmol/L of $AgNO_3$ to the basic induction medium, which significantly promoted the formation of embryoids in tetraploid and dihaploid anthers, as well as delayed the degree of anther browning. When *Liang et al. (2006)* studied the factors influencing potato anther culture, they found that the medium added with $AgNO_3$ (30 mg/L) and activated carbon (0.4%) could effectively reduce the degree of anther browning. Sucrose and potato extract, as carbon sources, were added to the anther induction medium to provide cell growth energy, maintain the osmotic pressure of the medium, and form the cytoskeleton. This has also been confirmed to be effective in reducing browning during anther culture. In this

study, the orthogonal design L9(3⁴) was adopted to evaluate the effect of four commonly used appendages on the induction rate of callus; the scheme and dosage of appendages with significant effects on anther induction were obtained.

## Ploidy identification and morphological observation

Theoretically, microspores contain only half the number of chromosomes in the somblastome tissue, and anther-cultured plantlets should be haploid (*Ahmadi & Ebrahimzadeh, 2020*; *Yuan et al., 2015*). However, the truth is that DHs (2N), triploid (3N), tetraploid (4N), and even higher ploidy levels occur due to the generation of spontaneous haploid genome replication, parent somatic tissue or different individuals without reduced gametes during anther culture (*Dunwell, 2010*; *Palmer, Keller & Arnison, 1996*; *Perera et al., 2008*). Therefore, it is essential to identify the doubling and purity of anther culture.

Flow cytometry has been widely used in plant ploidy identification by analyzing nuclear DNA content quickly and accurately (*Garcia-Fortea et al., 2021*; *Jia et al., 2014*). In this study, a flow cytometer was adopted to identify the chromosomal ploidy of anther culture plantlets. Regenerative plantlets with different ploidy levels, such as dihaploid, tetraploid, and octoploid, were obtained. We used FISH technology to distinguish the cell chromosomes at the root tip using a repeat sequence of 5S rDNA and 18S rDNA as probes during the metaphase of mitosis to identify polyploidy and aneuploidy more clearly. It was shown in the karyotypic analysis that 10 were dihaploid plantlets, indicating that they originated from pollen cells; 177 were tetraploid plantlets, which might originate from the somtrophic cell transformation, or 2n pollen directly formed by abnormal meiosis of anther, or natural chromosome doubling during the process of callus cell division; and 14 octoploid plantlets were aneuploidy, which might the doubling affected by the culture environment during the somatic embryogenesis.

Phenocycline characteristics are the main means of distinguishing between anther-cultured haploids, and parent-derived diploids (*Sharma, Sarkar & Pandey, 2010*). Phurejanot only classified potatoes from anther sources (*Pehu, Veilleux & Hilu, 1987*) but also compared male and female ligands to induce the formation of haploid morphological traits (*Lough, Varrieur & Veilleux, 2001*). In the study by *Sharma, Sarkar & Pandey (2010)*, no differences in the growth, color, quantity, and size of the small leaves and the color and size of the flower crown between the anther-induced haploids or tetraploids and its tetraploid donors were revealed. This might be because the anther donor was a self-pollination crop (*Watanabe et al., 1994*), resulting in high purity and low variation levels in its haploids. *Logue (1996)* believed that genetic changes in male ligand culture that produce variations included chromosomal aberrations in the number and structure, changes in nuclear DNA content, changes in nuclear gene sites, changes in reverse transcription transposure sequences, changes in mtDNA, and even epigenetic changes caused by DNA methylation (*Dogramaci-Altuntepe, Peterson & Jauhar, 2001*; *Muñoz Amatriaín et al., 2004*; *Törjék et al., 2001*; *Zagorska et al., 2004*).

SSR polymorphic markers have stability and co-dominance and are the primary means of identifying and distinguishing pure and hybrids. *Sharma, Sarkar & Pandey (2010)* used microsatellites to determine the phenotype variation of anther-cultured plantlets and the

polymorphism of nuclear microsatellites. Those researchers evaluated tetraploid potato anther-induced haploid genome changes and rearrangement. They proved that meiosis and allele mutations were the causes of variation in the culture of male ligands. Based on the identification results of the morphological and agronomic traits of induced plants from anther culture in this study, the morphological changes of diploids and agronomic traits were closely related to the changes in ploidy levels. Still, some anther-cultured plantlets and their donors showed significant differences in potato shape, the number of tubers, *etc.* Because of the differences between the phenomics and agronomic traits of anther-cultured plantlets and their receptors in this study, the next step will be to screen the potato SSR marker and identify the purity of the anther culture plants. With this, it will be possible to determine the source of these phenogram variations, thus providing technical support for creating the excellent new seed-quality of potatoes.

In conclusion, regenerative plantlets with improved agronomic characteristics were obtained from the anther of *S. tuberosum*. A relatively complete system for the anther culture in *S. tuberosum* was established, and varieties with high anther culture ability were screened out. The protocol presented here will benefit the creation of new potato species and the application of ploidy breeding in the future.

### Funding
This work was supported by the Agricultural Breeding Program of Ningxia Hui Autonomous Region (grant numbers 2019NYYZ01-2, 2014NYY201) and the Natural Science Foundation of Ningxia Hui Autonomous Region (grant number NZ17124, 2022AAC03428). The funders had no role in study design, data collection and analysis, decision to publish, or preparation of the manuscript.

### Grant Disclosures
The following grant information was disclosed by the authors:
Agricultural Breeding Program of Ningxia Hui Autonomous Region:  2019NYYZ01-2, 2014NYY201.
Natural Science Foundation of Ningxia Hui Autonomous Region:  NZ17124, 2022AAC03428.

### Competing Interests
The authors declare there are no competing interests.

### Author Contributions
- Li Zhang performed the experiments, analyzed the data, prepared figures and/or tables, authored or reviewed drafts of the article, and approved the final draft.
- Feng-jie Nie performed the experiments, prepared figures and/or tables, authored or reviewed drafts of the article, and approved the final draft.
- Lei Gong performed the experiments, prepared figures and/or tables, authored or reviewed drafts of the article, and approved the final draft.

- Xiao-yan Gan performed the experiments, prepared figures and/or tables, authored or reviewed drafts of the article, and approved the final draft.
- Guo-hui Zhang conceived and designed the experiments, authored or reviewed drafts of the article, and approved the final draft.
- Xuan Liu conceived and designed the experiments, authored or reviewed drafts of the article, supervision, and approved the final draft.
- Wen-jing Yang conceived and designed the experiments, authored or reviewed drafts of the article, supervision, and approved the final draft.
- Lei Shi conceived and designed the experiments, authored or reviewed drafts of the article, supervision, and approved the final draft.
- Yu-chao Chen conceived and designed the experiments, authored or reviewed drafts of the article, and approved the final draft.
- Rui-xia Xie conceived and designed the experiments, authored or reviewed drafts of the article, and approved the final draft.
- Zhi-qian Guo conceived and designed the experiments, authored or reviewed drafts of the article, and approved the final draft.
- Yuxia Song conceived and designed the experiments, authored or reviewed drafts of the article, and approved the final draft.

## Data Availability

The raw data is available in the Supplemental Files.

## Supplemental Information

Supplemental information for this article can be found online at http://dx.doi.org/10.7717/peerj.14984#supplemental-information.

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
