# Peer review of "Regenerative plantlets with improved agronomic characteristics caused by anther culture of tetraploid potato (Solanum tuberosum L.)"

_PeerJ, doi:10.7717/peerj.14984_

## Round 0.1 · original submission · Major Revisions

Dear Sir kindly revise your manuscript as suggested by the referees.

Reviewer 1 ·

Basic reporting

The article must be rewritten in scientific English and set up with more scientific rigor.
Literature references are sufficient.
The structure of the article must be improved.

Experimental design

The manuscript is interesting but needs improvement.

Validity of the findings

No comment

Reviewer 2 ·

Basic reporting

The authors provide an guidance for the potato ploidy breeding through exploring the qppopriate condition for anther culture in vitro and establishing an high efficient anther culture system of tetraploid potato. In general, this is a comprehensive study, the experiments were well conducted, and the presented results very convincing.However, there are yet many problems need to be addressed. In my opinion, the manuscript needs a major revision for reconsideration for possible publication in Peer J.

Experimental design

The basis for the design of experimental conditions should be listed in the text, such as the basis for choosing different hormone combinations.

Validity of the findings

the presented results very convincing

Additional comments

1. The english is poor, and there are many grammatical mistake in the text, I think it needs great improve.
2. Abreviations: the full name of the abreviations should be list for the first time it used in the manuscript,such as NAA, 2,4-D...including that in the abstract, please check this throughout the text.
3. There should be a space between the value and the unit.
4. The results should be rephrased, It's a little hard to read now.
5. The basis for the design of experimental conditions should be listed in the text, such as the basis for choosing different hormone combinations.
6. Suggest the Table1-table 3 transfered to the supplemental materials.
7. Most of the references were too old, and it is recommended to replace it with literatures with the recent 10 years.

·

Basic reporting

Dear authors

I carefully reviewed the manuscript entitled “”. At the present work, a highly efficient method to perform anther culture system of tetraploid potato was established, based in variations of hormones and culture composition. In my opinion, this work is very interesting because the induction of callus and obtention of plantlets of potato and other plants could means a form to filter pathogens like viruses and as in the present work shown also the plantlets could be increased in its characteristics.

I have several minor observations

1) For all the paper authors must separate the numerical values from the unities, in example: 3 mL; 5 d; 16 h; etc.
2) Line 24. Authors wrote: “and an high efficient”; there are a mistake in the redaction. I suggest change “an” per “a”.
3) The name of the potato species must be redacted properly as “Solanum tuberosum L.”; the letter L must be redacted without italics. Please check it for lines: 45, 335,
4) Line 47. Authors wrote “and strong tolerance to barren soil”. I suggest to change “barren” per “arid”.
5) Lines 59-60. Authors wrote “breeding efficiency. (Rokka, 2021; Song et al., 2005)”. The dot after the word “efficiently” must be eliminated, and a dot must be inserted after the cites.
6) Line 67. Authors wrote “S.chacoense Bitt and S.phureja Juz.”. A proper space must be inserted between the genus and species name (S. chacoense Bitt and S. phureja Juz.).
7) Lines 69-70. The phrase must be increased in quality “Homozygous bihaploids were obtained that were resistance to stem-nematode and antiviral through anther culture(Wenzel and Uhrig, 1981).” Additionally, a proper space must be inserted after the word “culture”.
8) Line 75. Authors wrote “regenerative plantlets(Rokka et al., 1996)”. A proper space must be inserted after the word “plantlets”.
9) Line 106. I suggest to change the word “inoculated” per “sown”. In the future, plantlets obtained using the present technology could be used to inoculate beneficial bacteria, therefore could be confusion between the words used. The same for lines 117 and 133.
10) Line 152. Authors wrote “About 1 cm2of leaf”. A proper space before the word “of” must be inserted.
11) Lines 184-185. Authors wrote “The 5SrDNA probe was plasmid pTa794 (Gerlach 185 and Dyer, 1980), and the 18S rDNA probe was plasmid pBR322 (Gerlach and Bedbrook, 1979)”. I do not understand the mean of the phrases. Maybe authors want to say that The 5SrDNA region inserted in the plasmid pTa794 was used as a probe, and the same for the other sentence. Please clarify.
12) The expression “in situ” must be redacted using italics. Please check the lines: 273, 280, 287, Figure 4 (three times).
13) Line 353. Authors wrote “promote the callus information”. The word “information” must be corrected to “formation”.
14) Line 355. Authors wrote “this part was not shown in this article”. In my opinion all the section “Pretreatment and potato anther culture” must be eliminated if authors do not present results and methodology.
15) Lines 444-445. To use italics for “Solanum tuberosum”.
16) Table 1. What does mean “Med iaN o.”? I suggest to write “Treatment”. Furthermore, in this table the abbreviations of the hormones must be defined.
17) Table 6. What does mean 13 and 14 in this table?
18) Figure 1. What does mean A, B, C and D? Define it in the legend.
19) Legend of figure 4. Authors wrote “of regenerated plant cells (G-L)” but must say “of regenerated plant cells (G-I”.
20) Legend of figure 6. Description of letter D is lost; I think it correspond to “F3-10 (diploid)”. Please clarify.
21) Figure 6. Photograph of letter “J” is lost. Please clarify.

Experimental design

In reference to the commentary 14 (14) Line 355. Authors wrote “this part was not shown in this article”. In my opinion all the section “Pretreatment and potato anther culture” must be eliminated if authors do not present results and methodology.) If aithors decide to mantain this secction then ins important describe the experimental process for it.

Validity of the findings

no comment

Additional comments

No additional comments

---

## Round 0.2 · Minor Revisions

Dear Sir

Kindly revise your manuscript as suggested

Reviewer 2 ·

Basic reporting

no comment

Experimental design

no comment

Validity of the findings

no comment

Additional comments

1. The word P-value in line 122, the "P" should be italic.
2. The symbol used should be consist throughout the MS, for example, Figure 4A-F in Line 278 and Figure 3A-D in

·

Basic reporting

Dear authors
All my questions and suggestions were considered in the new version.
Thanks a lot

Experimental design

no comment

Validity of the findings

no comment

Additional comments

no comment

---

## Round 0.3 · accepted · Accept

Authors have revised the manuscript and it can be accepted now.